# Fate of Surgical Patients with Small Nonfunctioning Pancreatic Neuroendocrine Tumors: An International Study Using Multi-Institutional Registries

**DOI:** 10.3390/cancers14041038

**Published:** 2022-02-18

**Authors:** In Woong Han, Jangho Park, Eun Young Park, So Jeong Yoon, Gang Jin, Dae Wook Hwang, Kuirong Jiang, Wooil Kwon, Xuefeng Xu, Jin Seok Heo, De-Liang Fu, Woo Jung Lee, Xueli Bai, Yoo-Seok Yoon, Yin-Mo Yang, Keun Soo Ahn, Chunhui Yuan, Hyeon Kook Lee, Bei Sun, Eun Kyu Park, Seung Eun Lee, Sunghwa Kang, Wenhui Lou, Sang-Jae Park

**Affiliations:** 1Division of Hepatobiliary and Pancreatic Surgery, Department of Surgery, Samsung Medical Center, School of Medicine, Sungkyunkwan University, 81 Irwon-ro, Gangnam-gu, Seoul 06351, Korea; cardioman76@gmail.com (I.W.H.); sojeong.yoon@samsung.com (S.J.Y.); jsheo@skku.edu (J.S.H.); 2Center for Liver and Pancreatobiliary Cancer, Research Institute and Hospital of National Cancer Center, Goyang 10408, Korea; khumedicaljang@hanmail.net; 3Biostatistics Collaboration Team, Research Institute and Hospital of National Cancer Center, 323 Ilsan-ro, Ilsandong-gu, Goyang-si 10408, Korea; 13140@ncc.re.kr; 4Department of Hepato-Biliary-Pancreatic Surgery, Changhai Hospital, Second Military Medical University, Shanghai 200433, China; jingang@smmu.edu.cn; 5Division of Hepatobiliary and Pancreatic Surgery, Department of Surgery, Asan Medical Center, University of Ulsan College of Medicine, Seoul 05505, Korea; dwhwang@amc.seoul.kr; 6Department of General Surgery, Pancreas Center, The First Affiliated Hospital, Nanjing Medical University, Nanjing 210029, China; jiangkuirong@163.com; 7Department of Surgery and Cancer Research Institute, Seoul National University College of Medicine, Seoul 03080, Korea; willdoc78@gmail.com; 8Department of Pancreatic Surgery, Zhongshan Hospital, Fudan University, Shanghai 200032, China; xu.xuefeng@zs-hospital.sh.cn; 9Department of Pancreatic Surgery, Huashan Hospital, Fudan University, Shanghai 200040, China; surgeonfu@163.com; 10Division of Hepatobiliary and Pancreatic Surgery, Department of Surgery, Yonsei University College of Medicine, Pancreatobiliary Cancer Center, Yonsei Cancer Center, Severance Hospital, Seoul 03722, Korea; wjlee@yuhs.ac; 11Department of Hepatobiliary and Pancreatic Surgery, The Second Affiliated Hospital, Zhejiang University, Hangzhou 310009, China; shirleybai@zju.edu.cn; 12Department of Surgery, Seoul National University Bundang Hospital, Seoul National University College of Medicine, Seoul 13620, Korea; yoonys@snubh.org; 13Department of General Surgery, The First Hospital of Peking University, Beijing 100034, China; yangyinmo@263.net; 14Department of Surgery, Keimyung University Dongsan Hospital, Keimyung University School of Medicine, Daegu 42601, Korea; ahnksmd@gmail.com; 15Department of General Surgery, The Third Hospital of Peking University, Beijing 100083, China; ychdoctor@163.com; 16Department of Surgery, Ewha Womans University College of Medicine, Seoul 07804, Korea; leehk@ewha.ac.kr; 17Department of Hepatobiliary and Pancreatic Surgery, The First Affiliated Hospital of Harbin Medical University, Harbin 150001, China; sunbei70@tom.com; 18Division of Hepatobiliary and Pancreatic Surgery, Department of General Surgery, Chonnam National University Hospital, Gwangju 61469, Korea; iameunkyu@gmail.com; 19Department of Surgery, Chung-Ang University Hospital, Chung-Ang University College of Medicine, Seoul 06973, Korea; selee508@cau.ac.kr; 20Division of Hepatobiliary and Pancreatic Surgery, Department of Surgery, Dong-A University Hospital, Busan 49201, Korea; kang3860@naver.com

**Keywords:** nonfunctioning neuroendocrine tumor of pancreas, prognosis, resection, risk factors

## Abstract

**Simple Summary:**

No consensus has been reached regarding whether nonmetastatic nonfunctioning neuroendocrine tumors of the pancreas (NF-pNETs) ≤ 2 cm should be resected or observed. In this retrospective international multicenter study, 483 patients who underwent resection for NF-pNETs ≤ 2 cm in 18 institutions from 2000 to 2017 were enrolled and their medical records were reviewed. Tumor size > 1.5 cm, Ki-67 index ≥ 3%, and nodal metastasis were independent adverse prognostic factors for survival after multivariable analysis. NF-pNET patients with tumors ≤ 1.5 cm can be observed if the preoperative Ki-67 index is under 3%, and if nodal metastasis is not suspected in preoperative radiologic studies. These findings support the clinical use to make decisions about small NF-pNETs.

**Abstract:**

Several treatment guidelines for sporadic, nonmetastatic nonfunctioning neuroendocrine tumors of the pancreas (NF-pNETs) have recommended resection, however, tumors ≤ 2 cm do not necessarily need surgery. This study aims to establish a surgical treatment plan for NF-pNETs ≤ 2 cm. From 2000 to 2017, 483 patients who underwent resection for NF-pNETs ≤ 2 cm in 18 institutions from Korea and China were enrolled and their medical records were reviewed. The median age was 56 (range 16–80) years. The 10-year overall survival rate (10Y-OS) and recurrence-free survival rate (10Y-RFS) were 89.8 and 93.1%, respectively. In multivariable analysis, tumor size (>1.5 cm; HR 4.28, 95% CI 1.80–10.18, *p* = 0.001) and nodal metastasis (HR 3.32, 95% CI 1.29–8.50, *p* = 0.013) were independent adverse prognostic factors for OS. Perineural invasion (HR 4.36, 95% CI 1.48–12.87, *p* = 0.008) and high Ki-67 index (≥3%; HR 9.06, 95% CI 3.01–27.30, *p* < 0.001) were independent prognostic factors for poor RFS. NF-pNETs ≤ 2 cm showed unfavorable prognosis after resection when the tumor was larger than 1.5 cm, Ki-67 index ≥ 3%, or nodal metastasis was present. NF-pNET patients with tumors ≤ 1.5 cm can be observed if the preoperative Ki-67 index is under 3%, and if nodal metastasis is not suspected in preoperative radiologic studies. These findings support the clinical use to make decisions about small NF-pNETs.

## 1. Introduction

Pancreatic neuroendocrine tumors (pNETs) account for less than 2% of all pancreatic cancers [1,2] and approximately 10% of tumors arising from the pancreas [3]. Their rare incidence and indolent biologic behavior with variable malignant potential has made it difficult to establish optimal management for pNETs [1,3,4]. Nonfunctioning neuroendocrine tumors of the pancreas (NF-pNET) account for 50–75% of all pNETs, and awareness of their incidence and prevalence in recent decades has increased due to the development of imaging technology and improved pathological diagnosis [1,2,5,6,7].

Several treatment guidelines for NF-pNETs have recommended resection, however, evidence is lacking for the best way to treat NF-pNETs ≤ 2 cm [8,9,10,11,12]. Some studies suggest that many small, asymptomatic pNETs are biologically indolent, do not enlarge or progress over time, show low nodal metastasis, and thus can be safely observed [13,14]. However, several reports emphasized that even small tumors can behave aggressively and that survival times improved after resection [15,16,17]. Currently, it is unclear how to preoperatively predict the malignant potential of small NF-pNETs, how to select patients for surgery, and how to determine the approach and extent of surgery that should be performed in patients selected for resection. 

The aims of this Korean–Chinese multi-institutional study are to analyze the postoperative outcomes and prognostic factors after resection of sporadic, nonmetastatic NF-pNETs ≤ 2 cm and to suggest surgical indications. 

## 2. Materials and Methods

### 2.1. Patients and Data Collection 

Patients with symptoms and biochemical evidence of excess pancreatic hormone are considered to have functioning pNETs, whereas patients with no symptoms, normal serum hormone levels, and no hereditary syndrome, such as MEN type I, are considered to have sporadic NF-pNETs [11]. Under Institutional Review Board approval (number: 2019-03-160-001), we retrospectively analyzed the clinicopathological variables of 329 Korean patients from 10 institutes participating in the Korean Tumor Registry System–Biliary Pancreas (KOTUS-BP) and 154 Chinese patients from 8 institutes. All institutions participating in this study are tertiary referral high-volume centers, and all patients were treated based on the guideline or consensus at the time. All participants underwent resection to treat sporadic, nonmetastatic NF-pNETs smaller than 2 cm, according to final pathologic reports between November 2000 and December 2017. When necessary, additional retrospective medical record review was performed. 

The variables collected for this work were age at diagnosis, sex, tumor size, location, nodal metastasis, perineural invasion, WHO 2010 tumor grade, mitotic count, Ki-67 index, and duration of follow-up. The OS time was defined as the time from the date of operation to the date of death or last known follow-up. The RFS time was measured from the date of operation until recurrence. The follow-up was updated in March 2020. In addition, information on surgery and complications was obtained for Korean patients. For these patients, postoperative complications were classified using the Clavien–Dindo classification. Parenchymal sparing resection (PSR) included central pancreatectomy, enucleation, and duodenal preserving pancreatic head resection (DPPHR), and standard resections involved PD or DP. MIS included laparoscopic or robotic resection of the pancreas. Major complications were defined as Clavien–Dindo grade III or higher. The POPF was defined using the 2016 International Study Group on Pancreatic Fistula definition [18].

### 2.2. Statistical Analysis

The baseline characteristics were summarized as median value (range) or frequency (percentage). Comparisons of variables and postoperative complications were performed using the Chi-squared test. OS and RFS rates were estimated using the Kaplan-Meier method, and the survival curves were presented with *p* of log-rank tests. The clinicopathological features associated with OS and RFS were analyzed using Cox proportional hazards models. Statistically significant variables in univariable analysis were included in multivariable analysis. The final model was determined using the backward selection method with elimination criterion of *p* > 0.05. Hazard ratios (HRs) and their 95% confidence intervals (CIs) were presented, and *p* less than 0.05 indicated statistical significance. All statistical analyses were performed using SAS version 9.4 (SAS Institute, Cary, NC, USA) and R project software (version 3.6.2).

## 3. Results

### 3.1. Clinicopathological Characteristics 

Table 1 provides the clinicopathologic details of the 483 patients. The median patient age was 56 (range 16–80) years with a male to female ratio of 1:1.44 (198:285). The tumors of 197 patients (40.8%) were in the pancreas head, and those of 286 patients were in the body or tail. The median tumor size was 1.4 cm (0.1–2.0 cm) and 24 patients (5.0%) had multiple tumors. Using the WHO 2010 grades (G), 364 patients (75.8%) had G1, 105 (21.9%) had G2, and 11 (2.3%) had G3. The numbers of patients with Ki-67 index <3%, 3–20%, and >20% were 388 (82.6%), 73 (15.5%), and 9 (1.9%), respectively. The numbers of patients with mitosis counts < 2, 2–20, and > 20 were 401 (91.1%), 39 (8.9%), and 0 (0%), respectively. Among 243 patients who had lymph node dissection or sampling, 32 patients (13.2%) had lymph node metastasis (LNM), and these patients accounted for 7.1% of all 483 patients (Table 1). 

### 3.2. Postoperative Complications According to Type of Surgery

Major complications occurred in 28 patients (8.5%), and the Postoperative pancreatic fistula (POPF; grade B or C) rate was 7.6% (*n* = 25) among the Korean patients. The postoperative outcomes of patients according to PSR or standard resection among the Korean patients are shown in Appendix A. No differences were observed in major complications, delayed gastric emptying, POPF, or postoperative hemorrhage. In addition, no statistically significant differences were found in any complication between patients who underwent MIS and those who underwent open resection among the Korean patients (Appendix A).

### 3.3. Survival and Risk Factor Analysis

The overall survival (OS) rate at 5 years and 10 years was 95.7 and 89.8%, respectively (Figure 1). The OS in patients with tumors ≤ 1.5 cm was more favorable than that in patients with tumors > 1.5 cm (*p* < 0.001, Figure 1b). However, the OS in patients with tumors ≤ 1.0 cm did not differ from those with tumors 1.0–1.5 cm (*p* = 0.511, Figure 1c). In multivariable analysis, older age (>65 years; HR 4.26, 95% CI 1.84–9.84, *p* = 0.001), tumor size (>1.5 cm; HR 4.28, 95% CI 1.80–10.18, *p* = 0.001), and LNM (HR 3.32, 95% CI 1.29–8.50, *p* = 0.013) were significant prognostic factors for OS. These results are summarized in Table 2.

The recurrence-free survival (RFS) rate at 5 years and 10 years was 95.7 and 93.1%, respectively (Figure 2). Total recurrence was identified in 21 patients (4.3%). The most common site of recurrence was the liver (*n* = 11), followed by the lymph nodes (*n* = 6). There were 7 local recurrences and 11 recurrences in multiple sites. RFS in patients with tumors ≤ 1.5 cm was more favorable than that in patients whose tumors were > 1.5 cm (*p* = 0.022, Figure 2b). In particular, the RFS in patients with LNM was poorer than in those without LNM (*p* < 0.001, Figure 2d). In the multivariable analysis, high Ki-67 index (≥3%; HR 9.06, 95% CI 3.01–27.30, *p* < 0.001), nodal metastasis (HR 3.68, 95% CI 1.22–11.11, *p* = 0.021), and perineural invasion (HR 4.36, 95% CI 1.48–12.87, *p* = 0.008) were independent prognostic factors for RFS. Table 3 summarizes these results. Additionally, the distribution of risk factors for OS and RFS is described in Appendix A. 

Further analysis was conducted on Korean patients for type of surgery and the OS of PSR showed no significant difference in prognosis compared to standard surgery, whereas MIS showed favorable prognosis compared to open resection (Appendix A). In addition, RFS did not differ significantly between PSR and standard resection, or between MIS and open resection in Korean patients (Appendix A).

## 4. Discussion

To the best of our knowledge, this study is the largest multi-institution surgical series from Korea and China (483 patients) to examine sporadic, nonmetastatic NF-pNETs ≤ 2 cm, and thus it provides better external validity than small single-center cohorts. In this study, tumor size (>1.5 cm; HR 4.28, 95% CI 1.80–10.18, *p* = 0.001) and nodal metastasis (HR 3.32, 95% CI 1.29–8.50, *p* = 0.013) were independent adverse prognostic factors for OS (Table 2). Additionally, perineural invasion (HR 4.36, 95% CI 1.48–12.87, *p* = 0.008) and high Ki-67 index (≥3%; HR 9.06, 95% CI 3.01–27.30, *p* < 0.001) were independent prognostic factors for poor RFS (Table 3). 

Although surgery used to be the cornerstone of management for small NF-pNETs, that practice has been recently challenged. In view of the severe and frequent complications following pancreatic surgery and the natural history of sporadic NF-pNET smaller than 2 cm, observation without resection has recently been proposed as a possible option. To date, tumor size has been the main determinant when deciding on an operative or observational policy [7,10,11,12,13,17,19]. In several recently published guidelines [10,11,12], there is no agreement on whether to perform surgery or initial observation for pNETs ≤ 2 cm. One of the remarkable results of this study is the finding that the size classification criteria should be changed from 1 or 2 cm to 1.5 cm, as our results indicate that NF-pNETs ≤ 1.5 cm have a better prognosis than those larger than 1.5 cm (Table 2). NF-pNETs ≤ 1.5 cm can be observed because their low risk counterbalances the potential for morbidity, mortality, and exocrine and endocrine deficiencies associated with pancreatic resection. Therefore, changing the surgical indication from 1–2 cm to 1.5 cm is reasonable.

Other information to consider when choosing surgical candidates with pNETs smaller than 2 cm is the rate of nodal metastases. Nodal metastasis predicts poor prognosis [5,11,14,20,21,22]. In this study, LNM was identified in 32 (7.1%) of all 483 patients and was an independent risk factor for poor OS and RFS (Table 2 and Table 3). In addition, the rate of nodal metastasis in patients with NF-pNETs smaller than 1.5 cm was 5.5% (Appendix A). Several previous reports warned that small pNETs, regardless of location, had a risk of LNM from 12.9% to 27.3% [11,14,21,22]. Although we found that the risk of LNM in patients with small NF-pNETs was lower than in previous studies, we did still find a considerable risk. Thus, standard nodal dissection including suspicious metastatic nodes in preoperative imaging is warranted, and LN sampling can be considered if imaging is negative.

With small NF-pNETs, preoperative endoscopic ultrasound (EUS)-fine needle biopsy (FNB) to determine the Ki-67 level as a predictor of malignancy assists in decision making, because tumor grading has clear implications in terms of prognosis [23,24]. In this study, tumor grading was one of the risk factors for RFS (Table 3). A recent study of data for 210 patients from 16 European centers concluded that patients with grade 2 or 3 tumors, which were independent risk factors for poor disease-free survival, should undergo resection, whereas patients with pNETs smaller than 2 cm, could reasonably be managed with surveillance [7]. Therefore, EUS-FNB should be considered for patients for whom surgical indications are questionable [11,25].

As in previous studies [23,26], vascular invasion and perineural invasion were adverse risk factors for survival (Table 2 and Table 3). Although it is difficult to determine in preoperative imaging whether vascular or perineural invasions is present, they are clearly factors to be considered before surgery. An age older than 65 years was an adverse prognostic factor for OS (Table 2), so it should play a role in selecting patients for surgery versus surveillance. However, older patients also have a higher risk of mortality from surgery, a higher likelihood of comorbidities, a shorter life expectancy, and shorter surveillance time compared with younger patients. As a result, caution should be taken in interpreting these contradictory results as a uniform endorsement of surgical resection in older patients with small pNETs who will potentially benefit from surgical resection.

Patients with NF- pNETs smaller than 2 cm have excellent long-term survival, which makes it important to optimize their quality of life in terms of pancreatic function following surgical intervention. A pancreaticoduodenectomy (PD) is the gold standard for lesions of the pancreatic head, whereas a distal pancreatectomy (DP) is used for tumors in the body and tail. However, these surgical procedures are associated with a substantial loss of functional pancreatic and extrapancreatic tissues. In addition, standard pancreatic resections, including multiorgan resection, have considerable postoperative morbidity, a substantial risk of mortality, and inevitable long-term functional impairments [27,28], and no data supports that an aggressive resection to obtain wide surgical margins is justified for pNETs [11]. PSR, including central pancretectomy, enucleation, and DPPHR, have been advocated in select pNET patients to minimize morbidity and maintain pancreatic endocrine and exocrine function [9,27,29]. In this study, all variables regarding complications and survival did not show statistically significant differences between PRS and standard resection among the Korean patients (Appendix A). As mentioned above, accurate assessment of LNM is important to predict prognosis by staging, and PRS generally has a low lymph node yield rate. In several previous studies, the rate of no lymph node sampling was higher in patients undergoing PRS than in patients who underwent standard operations [5,11,30,31]. Patients with low-risk NF-pNETs ≤ 1.5 cm, who are predicted to have long survival times, and those who develop pNETs at a young age have the most to gain from preserved pancreatic function and can thus be considered most appropriate for planned PRS with selective lymphadenectomy instead of a standard resection. Whether to proceed with PRS or a standard resection is a delicate decision, that should be discussed and preferably made with the patients.

As in this study (Appendix A), other studies have reported favorable results from minimally invasive surgery (MIS) compared with conventional open pancreatectomy [32,33,34,35,36]. Several guidelines have considered different approaches for resecting pNETs, especially those in the tail of the pancreas, and laparoscopic or robotic DP has been considered to be safe and effective with satisfactory postoperative and oncologic outcomes [8,11,35,37]. Unlike with DP, results with MIS PD might not be as favorable as those from the conventional open approach [38,39]. Therefore, left-sided pancreatectomy should consider a laparoscopic or robotic approach first, but minimally invasive PD should be approached carefully considering the surgeon’s experience or the condition of the patient. 

The present study has several limitations. First, we only had access to data on patients who underwent resection, and how many patients were under surveillance in the participating centers during the same period is unknown. If long-term RFS after curative surgery is excellent, the results of non-operative management are of prime interest. Second, this study did not have data about the size discrepancy between preoperative CT images and pathologic reports. Sallinen et al. [7] reported that significant discrepancies of those measurements should be noted. If possible, this limitation could have been reduced if the imaging and pathology data of Korean and Chinese patients were reviewed once again centrally. However, due to the nature of the retrospective study, it was not possible to directly transfer image and pathology data between countries. In addition, we could obtain information about the type of surgery or surgical complications only from the Korea multicenter database. If that information could have been included in the multivariable analysis process, more sophisticated and reliable statistical analysis would have been possible. Last, selection bias could have affected the retrospective analysis. Patient cohorts of this study were inconsistent, including surgical indication and inconsistent surgical methods between institutions, due to the lack of a unified management strategy for small pNETs. 

## 5. Conclusions

NF-pNETs smaller than 2 cm showed considerable recurrence after resection when the tumor was larger than 1.5 cm, Ki-67 index was over 3%, or nodal metastasis was present. NF-pNET patients with tumors ≤ 1.5 cm can be observed without resection if the preoperative Ki-67 index is low, assuming preoperative tissue diagnosis is possible, and nodal metastasis is not suspected in preoperative radiologic studies. Therefore, the proposed surgical indication is expected to help stratify the patient’s prognosis and provide comprehensive clinical decision-making.

## Figures and Tables

**Figure 1 cancers-14-01038-f001:**
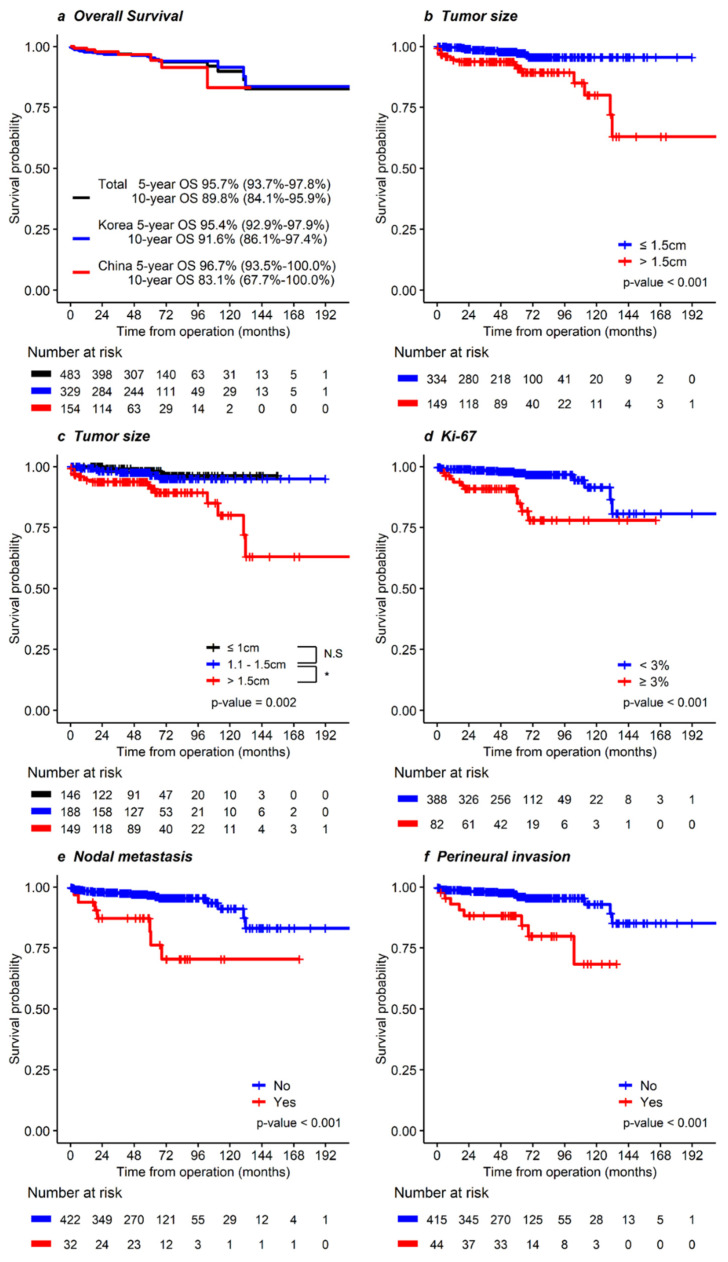
Kaplan-Meier curves for overall survival in patients with NF-pNETs ≤ 2 cm. In (**a**) total, Korean, and Chinese patients and according to (**b**) tumor size ≤ 1.5 or >1.5 cm, (**c**) tumor size ≤ 1, 1–1.5, or >1.5 cm, (**d**) Ki-67 index <3, or ≥3%, (**e**) nodal metastasis, and (**f**) perineural invasion.

**Figure 2 cancers-14-01038-f002:**
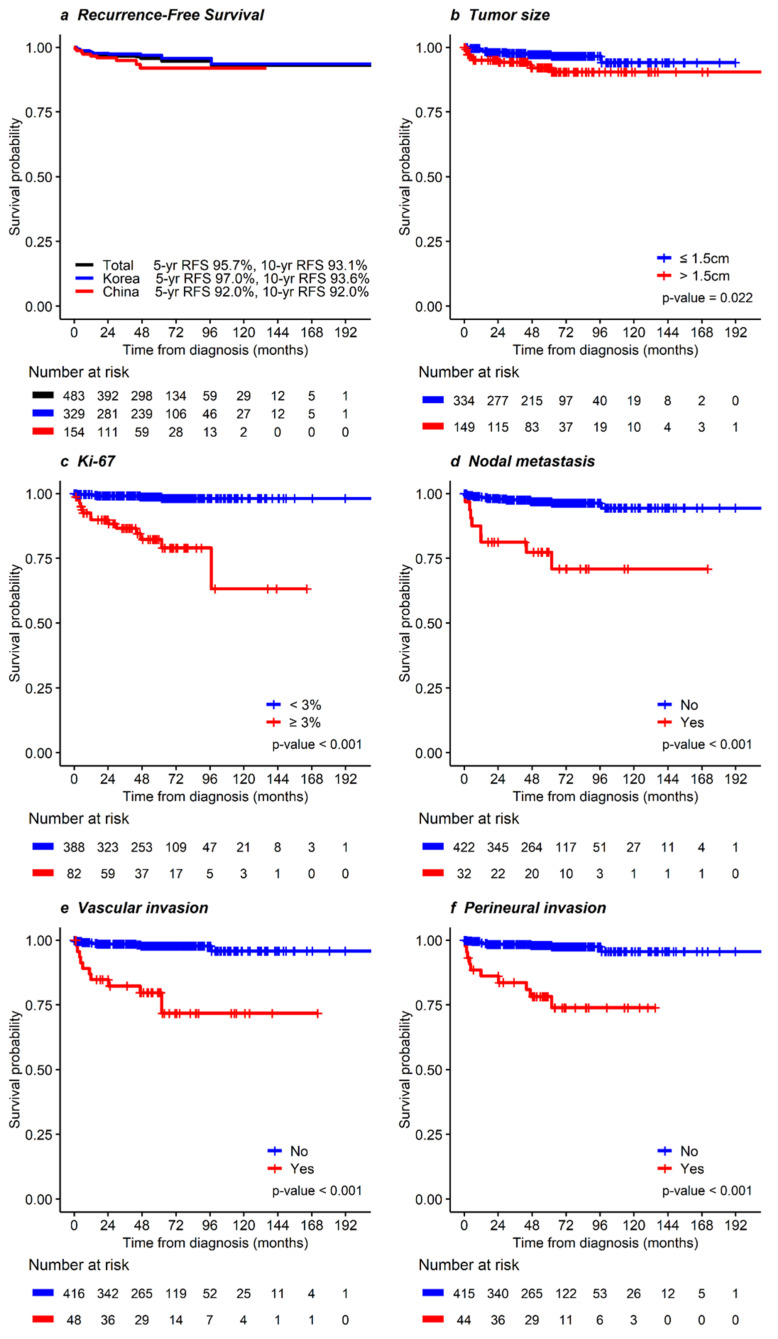
Kaplan-Meier curves for recurrence-free survival in patients with NF-pNET ≤ 2 cm. In (**a**) total, Korean, and Chinese patients and according to (**b**) tumor size ≤ 1.5 or > 1.5 cm, (**c**) Ki-67 index < 3, or ≥ 3%, (**d**) nodal metastasis, (**e**) vascular invasion, and (**f**) perineural invasion.

**Table 1 cancers-14-01038-t001:** Clinical characteristics of the patients.

Characteristics	Total (*n* = 483)	Korea (*n* = 329)	China (*n* = 154)
Sex (male/female)	198/285 (1:1.44)	134/195 (1:1.46)	64/90 (1:1.41)
Age (median, range)	56.0 (16–80)	56.0 (17–77)	55.0 (16–80)
Tumor size (median, range) (cm)	1.4 (0.1–2.0)	1.4 (0.1–2.0)	1.5 (0.1–2.0)
Tumor size (≤1/1.1–1.5/>1.5–2 cm) (*n*, %)	146 (30.2%)/188 (38.9%)/149 (30.8%)	95 (28.9%)/131 (39.8%)/103 (31.3%)	51 (33.1%)/57 (37.0%)/46 (29.9%)
Number of tumors (1/>1) (*n*, %)	459 (95.0%)/24 (5.0%)	316 (96.0%)/13 (4.0%)	143 (92.9%)/11 (7.1%)
Tumor location (head/elsewhere) (*n*, %)	197 (40.8%)/286 (59.2%)	137 (41.6%)/192 (58.4%)	60 (39.0%)/94 (61.0%)
WHO 2010 grade (1/2/3) (*n*, %)	364 (75.8%)/105 (21.9%)/11 (2%)	259 (78.7%)/66 (20.1%)/4 (1.2%)	105 (68.2%)/39 (25.3%)/7 (4.5%)
Ki-67 index (median, range) (%)	1.0 (0–80)	1.0 (0–80)	2.0 (0–60)
Ki-67 index within WHO 2010 grade (G1/G2/G3) (median, range)	1 (0–2.5)/3.7 (0–20)/35 (2–80)	1 (0–2.5)/3.05 (0–10)/61.685 (2–80)	1 (0–2)/4 (1–20)/30 (3–60)
Ki-67 index (<3/3–20/>20) (*n*, %)	388 (82.6%)/73 (15.5%)/9 (1.9%)	282 (85.7%)/34 (10.3%)/3(1.0%)	106 (68.8%)/39 (25.3%)/6 (3.9%)
Mitotic count (median, range)	1 (0–20)	1 (0–20)	1 (0–20)
Mitotic count (<2/≥2) (*n*, %)	401 (91.1%)/39 (8.9%)	284 (90.5%)/30 (9.6%)	106 (70.2%)/45 (29.8%)
Nodal dissection (*n*, %)	243 (50.3%)	164 (49.8%)	79 (51.3)
Nodal metastasis (*n*, %)	32 (7.1%)	15 (4.6%)	17 (13.6%)
Tumor margin (+) (*n*, %)	27 (5.7%)	24 (7.3%)	3 (2%)
Adjacent organ invasion (*n*, %)	9 (1.9%)	2 (0.6%)	7 (4.5%)
Vascular invasion (*n*, %)	48 (10.3%)	38 (11.6%)	10 (7.4%)
Perineural invasion (*n*, %)	44 (9.6%)	27 (8.2%)	17 (13.1%)

**Table 2 cancers-14-01038-t002:** Risk factor analysis for overall survival.

Variables	Univariable Analysis		Multivariable Analysis	
	HR (95% CI)	*p*-Value	HR (95% CI)	*p*-Value
Older age (>65 years)	5.14 (2.31–11.41)	<0.001	4.26 (1.84–9.84)	0.001
Tumor size (>1.5 cm)	3.98 (1.75–9.01)	0.001	4.28 (1.80–10.18)	0.001
Tumor size				
≤1 cm	1	(0.004)		
1.1–1.5 cm	1.59 (0.40–6.37)	0.511		
>1.5 cm	5.28 (1.54–18.15)	0.008		
WHO grade 2010				
G1	1	(<0.001)	-	-
G2	1.25 (0.44–3.51)	0.676	-	-
G3	31.64 (12.18–82.19)	<0.001	-	-
Ki-67 index (%)				
<3	1			
≥3	4.62 (2.07–10.35)	<0.001		
Mitotic count/HPF (≥2)	2.11 (0.61–7.30)	0.240		
Nodal metastasis	5.14 (2.14–12.34)	<0.001	3.32 (1.29–8.50)	0.013
Positive resection margin	3.28 (1.12–9.62)	0.031	4.30 (1.36–13.58)	0.013
Vascular invasion	5.17 (2.32–11.56)	<0.001		
Perineural invasion	4.73 (2.01–11.11)	<0.001		

HR: hazard ratio; CI: confidence interval; HPF: high power field; Multivariable analysis included Ki-67 and mitotic count instead of WHO grade 2010.

**Table 3 cancers-14-01038-t003:** Risk factor analysis for recurrence-free survival.

Variables	Univariable Analysis		Multivariable Analysis	
	HR (95% CI)	*p*-Value	HR (95% CI)	*p*-Value
Tumorsize (>1.5cm)	2.63 (1.12–6.19)	0.027		
Tumorsize				
≤1cm	1	(0.072)		
1.1–1.5cm	1.86 (0.48–7.20)	0.369		
>1.5cm	3.88 (1.08–13.92)	0.037		
WHOgrade2010				
G1	1	(<0.001)	-	-
G2	6.53 (2.19–19.50)	0.001	-	-
G3	91.30 (28.32–294.32)	<0.001	-	-
Ki-67index (%)				
<3	1		1	
≥3	15.77 (5.66–43.94)	<0.001	9.06 (3.01–27.30)	<0.001
Mitoticcount/HPF (≥2)	2.96 (0.81–10.75)	0.100		
Nodalmetastasis	8.62 (3.57–20.80)	<0.001	3.68 (1.22–11.11)	0.021
Positiveresectionmargin	1.96 (0.45–8.47)	0.370		
Adjacentorganinvasion	11.38 (3.31–39.12)	<0.001		
Vascularinvasion	11.07 (4.59–26.74)	<0.001		
Perineuralinvasion	10.85 (4.41–26.71)	<0.001	4.36 (1.48–12.87)	0.008

HR: hazard ratio; CI: confidence interval; HPF: high power field; Multivariable analysis included Ki-67 and mitotic count instead of WHO grade 2010.

## Data Availability

The data presented in this study are available in this article (and Appendix A).

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
