# Peer review of "Fate of Surgical Patients with Small Nonfunctioning Pancreatic Neuroendocrine Tumors: An International Study Using Multi-Institutional Registries"

_cancers, 2022, doi:10.3390/cancers14041038_

Round 1
Reviewer 1 Report
The Authors present a retrospective multicenter series on sporadic, non-functioning pNETs <2cm with the aims to evaluate surgical outcomes and overall prognosis and to suggest specific surgical indications . In my view, the manuscript could be of interest to the clinical NET community as it summarizes valuable information, although a control group of patients treated non-surgically is greatly missing. Overall, I believe the paper could be of interest to the readership of Cancers. However, there are some major points that need to be addressed that could improve the quality of the paper. The manuscript would benefit from major revision and clarification with respect to its limitations.
Major points:
- The cohort of pNET patients included in the present study need to be better characterized in both the Methos and Results section. Please clarify whether tumor size was determined preoperative on CT, MRI or EUS or postoperatively according to the final pathology report. Did the Authors only include well differentiated tumors? Please clarify how many patient were subjected to EUS-guided biopsies and the Ki67 was known before the operation. Finally, did the Authors include patients with small primaries and evident nodal metastases on cross-sectional imaging or EUS prior to surgery? The later needs further clarification and if so, these patients should be excluded from the analysis.
- Please use standard nomenclature folowing the STROBE guideline to report the study results. In particular, please provide hazard ratios in the the abstract, as well as the OS and RFS figures. Please define a minimum number of subjects remaining at risk after which Kaplan-Meier survival plots for time-to-event outcomes should be curtailed, as, once the number remaining at risk drops below this minimum, the survival estimates is no longer meaningful in the context of the investigation. Please provide apart from log-rank p-value also 95%CI on the figures.
- Although OS is the golden standard to present oncological outcomes, could the authors present Disease-specific Survival outcomes? This would be particularly meaningfull for older patients with small tumors included in the study to address if there is any benefit of surgery in this setting.
- How did the Authors choose tumor size cut-off 1,5 cm and Ki67 cut-off 3%? It would be more appropriate to conduct a ROC analysis in their dataset to assess best potential cut-offs as predictors of recurrence and/or death.
- One important flaw of the study in my view is that the Authors do not provide the results of a multivariable Cox-regression model for OS and RFS including more variables: age, gender, tumor size, primary tumor location, Ki67 index, type of surgery, vascular and perineural invasion to make a clinical inference on significant predictors for death and recurrence. This is very important as the univariable analyses peresented in the present study is largely subjected to selection bias and several factors affect each other.
- In table 1, please provide Ki67 median and range within each grade (G1,G2, G3).
- Please summarize the main findings of your study in the first paragraph of the discussion.
- Discussion: lines 185-187. This part is contradicting. Please consider if it could be rewritten.
- Please consider to better highlight the limitations of the present study with a lack of central re-review for histopathological specimens and also a lack of central radiological review. In additition, selection bias along with a potential ambiquity over the criteria to select patients for surgery.
Reviewer 2 Report
Notes to authors
The present study aimed to study whether nonmetastatic nonfunctioning neuroendocrine tumors of the pancreas (NF-pNETs) ≤ 2 cm should be resected or observed. In this retrospective international multicenter study, 483 patients who underwent resection for NF-46 pNETs ≤ 2cm in 18 institutions from 2000 to 2017 were enrolled and their medical records were reviewed. The authors observed that tumor size > 1.5cm, Ki-67 index ≥ 3%, and nodal metastasis were independent adverse prognostic factors for survival after multivariable analysis. The authors thus concluded that NF-pNET patients with tumors ≤ 1.5cm can be observed if the preoperative Ki-67 index is under 3%, and if nodal metastasis is not suspected in preoperative radiologic studies.
This is a well-written retrospective study, arising from two expert groups, with an adequate methodology to reduce bias.
Some minor corrections should be made.
Comments
Explain abbreviations at first appearance (line 103: PSR, DPPHR)
Round 2
Reviewer 1 Report
The Authors have addressed the Reviewers' concerns point by point and revised the manuscript adequately, improving the overall quality of the paper.
This manuscript is a resubmission of an earlier submission. The following is a list of the peer review reports and author responses from that submission.